# Learning Tree Structured Potential Games

**Vikas K. Garg**
CSAIL, MIT
vgarg@csail.mit.edu

**Tommi Jaakkola**
CSAIL, MIT
tommi@csail.mit.edu

## Abstract

Many real phenomena, including behaviors, involve strategic interactions that can be learned from data. We focus on learning tree structured potential games where equilibria are represented by local maxima of an underlying potential function. We cast the learning problem within a max margin setting and show that the problem is NP-hard even when the strategic interactions form a tree. We develop a variant of dual decomposition to estimate the underlying game and demonstrate with synthetic and real decision/voting data that the game theoretic perspective (carving out local maxima) enables meaningful recovery.

## 1 Introduction

Structured prediction methods [1; 2; 3; 4; 5] are widely adopted techniques for learning mappings between context descriptions $x \in \mathcal{X}$ and configurations $y \in \mathcal{Y}$. The variables specifying each configuration $y$ (e.g., arcs in natural language parsing) are typically mutually dependent and it is therefore beneficial to predict them jointly rather than individually. The predicted $y$ often arises as the highest scoring configuration with respect to a parameterized scoring function that decomposes into terms that couple two or more variables together to model their interactions. Structured prediction methods have been broadly useful across areas, from computational biology (e.g., molecular arrangements, alignments), natural language processing (e.g., parsing, tagging), computer vision (e.g., segmentation, matching), and many others. However, the setting is less suitable for modeling strategic interactions that are better characterized in terms of local consistency constraints.

We consider the problem of predicting configurations $y$ that represent game theoretic equilibria. Such configurations are unlikely to coincide with the maximum of a global scoring function as in structured prediction. Indeed, there may be many possible equilibria in a specific context, and the particular choice may vary considerably. Each possible configuration is nevertheless characterized by local constraints that represent myopic optimizations of individual players. For example, senators can be thought to vote relative to give and take deals with other closely associated senators. Several assumptions are necessary to make the game theoretic setting feasible.

We abstract the setting as a *potential game* [6; 7; 8] among the players, and define a stochastic process to model the dynamics of the game. A game is said to be a potential game if the incentive of all players to change their strategy can be expressed using a single global *potential function*. Every potential game is guaranteed to have at least one (possibly multiple) pure strategy Nash equilibria [9], and we will exploit this property in modeling and analyzing several real world scenarios. Note that each pure Nash equilibrium corresponds to a local optimum of the underlying potential function rather than the global optimum as in structured prediction.

We further restrict the setting by permitting the payoff of each player to depend only on their own action and the actions of their neighbors (a subset of the other players). Thus, we may view our setting as a graphical game [10; 11]. In this work, we investigate potential games where the graphical structure of the interactions form a tree. The goal is to recover the tree structured potential function that supports observed configurations of actions as locally optimal solutions. We prove that it is

NP-hard to recover such games under a max-margin setting. We then propose a variant of dual decomposition (cf. [12; 13]) to learn the tree structure and the associated parameters.

## 2 Setting

We commence with the game theoretic setting. There are $n$ players indexed by a position in $[n] \triangleq \{1, 2, \ldots, n\}$. These players can be visualized as nodes of a tree-structured graph $T$ with undirected edges $E$. We denote the set of neighbors of node $i$ by $N_i$, i.e., $(i, j) \in E \iff j \in N_i \land i \in N_j$, and abbreviate $(i, j) \in E$ as $ij \in T$ without introducing ambiguity. Each player $i$ has a finite discrete set of strategies $\mathcal{Y}_i$. A strategy profile or label configuration is an $n$-dimensional vector of the form $y = (y_1, y_2, \ldots, y_n) \in \mathcal{Y} = \prod_{i=1}^{n} \mathcal{Y}_i$. We denote the parametric potential function associated with the tree by $f(y; x, T, \theta)$, where $y$ is a strategy profile, $\theta$ the set of parameters, and $x \in \mathcal{X}$ is a context [14]. We obtain an $(n - 1)$-dimensional vector $y_{-i} = (y_1, \ldots, y_{i-1}, y_{i+1}, \ldots, y_n)$ by considering the strategies of all players other than $i$. Thus, we may equivalently write $y = (y_i, y_{-i})$. Moreover, we use $y_{N_i}$ to denote the strategy profile pertaining to the neighbors of node $i$. We can extract from $f(y; x, T, \theta)$ individual payoff (or cost) functions $f_i(y_i, y_{N_i}; x, T, \theta), i \in [n]$, which merely include all the terms that pertain to the strategy of the $i^{th}$ player $y_i$.

The choice of a particular equilibrium (local optimum) in a context results from a stochastic process. Starting with an initial configuration $y$ at time $t = 0$ (e.g., chosen at random), the game proceeds in an iterative fashion: during each subsequent iteration $t = 1, 2, \ldots$, a player $p_t \in [n]$ is chosen uniformly at random. The player $p_t$ then computes the best response candidate set

$$Z_{p_t} = \arg \max_{z \in \mathcal{Y}_{p_t}} f_{p_t}(z, y_{N_{p_t}}; x, T, \theta),$$

and switches to a strategy within this set uniformly at random if their current strategy does not already belong to this set, i.e., player changes their strategy only if a better option presents itself. The game finishes when a locally optimal configuration $\hat{y} \in \mathcal{Y}$ has been reached, i.e., when no player can improve their payoff unilaterally. Since many locally optimal configurations could have been reached in the given context $x$, the stochastic process induces a distribution over the strategy profiles.

We assume that our training data $S = \{(x^1, y^1), \ldots, (x^M, y^M)\}$ is generated by some distribution over contexts and the induced conditional distribution over strategy profiles with respect to some tree structured potential function. Our objective is to learn both the underlying tree structure $T$ and the parameters $\theta$ using a max-margin setting. Specifically, given $S$, we are interested in finding $T$ and $\theta$ such that

$$\forall \, m \in [M], i \in [n], y_i \in \mathcal{Y}_i, f(y^m; x^m, T, \theta) \geq f(y_{-i}^m, y_i; x^m, T, \theta) + e(y, y^m),$$

where $e(y, y^m)$ is a non-negative loss (e.g. Hamming loss), which is 0 if and only if $y = y^m$. Note that the maximum margin framework does not make an explicit use of the assumed induced distribution over equilibria.

The setting here is superficially similar to relaxations of structured prediction tasks such as pseudo-likelihood [15] or decomposed learning [16]. These methods are, however, designed to provide computationally efficient approximations of the original structured prediction task by using fewer constraints during learning. Instead, we are specifically interested in modeling the observations as locally optimal solutions with respect to the potential function.

We only state the results of our theorems in the main text, and defer all the proofs to the Supplementary.

## 3 Learning Tree Structured Potential Games

We first show that it is NP-hard to learn a tree structured potential game in a discriminative max-margin setting. Previous hardness results are available about learning structured prediction models under global constraints and arbitrary graphs [15], and under global constraints and tree structured models [17], also in a max-margin setting.

**Theorem 1.** Given a set of training examples $S = \{(x^m, y^m)\}_{m=1}^{M}$ and a family of potential functions of the form

$$f(y; x, T, \theta) = \sum_{ij \in T} \theta_{ij}(y_i, y_j) + \sum_{i} \theta_i(y_i) + \sum_{i} x_i(y_i),$$

it is NP-hard to decide whether there exists a tree $T$ and parameters $\theta$ (up to model equivalence) such that the following holds:

$$\forall \, m, i, y_i, f(y^m; x^m, T, \theta) \geq f(y^m_{-i}, y_i; x^m, T, \theta) + e(y, y^m).$$

### 3.1 Dual decomposition algorithm

The remainder of this section concerns with developing an approximate method for learning the potential function by appeal to dual decomposition. Dual decomposition methods are typically employed to solve inference tasks over combinatorial structures (e.g., [12; 13]). In contrast, we decompose the problem on two levels. On one hand, we break the problem into independent local neighborhood choices and use dual variables to reconcile these choices across the players so as to obtain a single tree-structured model. On the other hand, we ensure that initially disjoint parameters mediating the interactions between a player and its neighbors are in agreement across the edges in the resulting structure. The two constraints ensure that there is a single tree-structured global potential function.

For each node $i$, let $N_i$ be the set of neighbors of $i$ represented in terms of indicator variables such that $N_{ij} = 1$ if $i$ selects $j$ as a neighbor. $N_{ij}$ can be chosen independently from $N_{ji}$ but the two will be enforced to agree at the solution. We will use $N_i$ as a set of neighbors and as a set of indicator variables interchangeably. Similarly, we decompose the parameters into node potentials $\theta_i \cdot \phi(y_i; x) = \theta_i(y_i; x)$ and edge potentials $\theta_{ij} \cdot \phi(y_i, y_j; x) = \theta_{i,j}(y_i, y_j; x)$ where again $\theta_{ij}$ may be chosen separately from $\theta_{ji}$ but will be encouraged to agree. The set of parameters associated with each player then consists of locally controllable parameters $\Theta_i = \{\theta_i, \theta_{i\cdot}\}$ and $N_i$, where $N_i$ selects the relevant subset of interaction terms:

$$f(y; x, N_i, \Theta_i) = \theta_i(y_i; x) + \sum_{j \neq i} N_{ij} \theta_{i,j}(y_i, y_j; x)$$

Given a training set $S = \{(x^1, y^1), \ldots, (x^M, y^M)\}$, the goal is to learn the set of neighbors $\mathcal{N} = \{N_1, \ldots, N_n\}$, and weights $\Theta = \{\Theta_1, \ldots, \Theta_n\}$ so as to minimize

$$\frac{1}{2}||\Theta||^2 + \frac{C}{Mn} \sum_{i=1}^{n} \sum_{m=1}^{M} \underbrace{\max_{y_i} \left[ f(y^m_{-i}, y_i; x^m, N_i, \Theta_i) - f(y^m; x^m, N_i, \Theta_i) + e(y_i, y^m_i) \right]}_{\triangleq R_{mi}(N_i, \Theta_i)} \quad (1)$$

subject to $\mathcal{N}$ forming a tree and $\Theta$ agreeing across the players. Let $R_i(N_i, \Theta_i) = C/(Mn) \sum_m R_{mi}(N_i, \Theta_i)$. We force the neighbor choices to agree with a global tree structure represented by indicators $\mathcal{N}'$. Similarly, we enforce parameters $\Theta_i$ to agree across neighbors. The resulting Lagrangian can be written as

$$\sum_{i=1}^{n} \underbrace{\left[ \frac{1}{2}||\Theta_i||^2 + R_i(N_i, \Theta_i) + \sum_{j \neq i}(\delta_{ij} N_{ij} + \lambda_{ij} \cdot \theta_{ij}) \right]}_{L(\Theta_i, N_i; \delta, \lambda)} + \underbrace{\left[ -\sum_{i, j \neq i} \delta_{ij} N'_{ij} + G(\mathcal{N}') \right]}_{G(\mathcal{N}', \delta)}$$

where $G(\mathcal{N}') = 0$ if $\mathcal{N}'$ forms a tree and $\infty$ otherwise, and $\lambda_{ij} = -\lambda_{ji}$. For the dual decomposition algorithm, we must be able to solve $\min_{\Theta_i} L(\Theta_i, N_i; \delta, \lambda)$ to obtain $\Theta_i^*$ and $\min_{N_i} L(\Theta_i, N_i; \delta, \lambda)$ to get $N_i^*$. The former is a QP while the latter is more challenging though may permit efficient solutions via additional relaxations, exploiting combinatorial properties in restricted cases (sub-modularity), or even brute force for smaller problems. $G(\mathcal{N}', \delta)$ corresponds to a minimum weighted spanning tree, and thus can be efficiently solved using any standard algorithm like Borůvka's, Kruskal's or Prim's.

The basic dual decomposition alternatively solves $\Theta_i^*$, $N_i^*$, and $\mathcal{N}'^*$, resulting in updates of the dual variables based on disagreements. While the method has been successful for enforcing structural constraints (e.g., parsing), it is less appropriate for constraints involving continuous variables. To address this, we employ the alternating direction method of multipliers (ADMM) [18; 19; 20] for parameter agreements. Specifically, we encourage $\theta_{i\cdot}$ and $\theta_{\cdot i}$ to agree with their mean $u_{i\cdot}$, by introducing an additional term to the Lagrangian $L$

$$L_A(\Theta_i, N_i; u_{i\cdot}, \delta, \lambda) = L(\Theta_i, N_i; \delta, \lambda) + \frac{\rho}{2}||\theta_{i\cdot} - u_{i\cdot}||^2$$

where $u_{i.}$ is updated as an independent parameter.

There are many ways to schedule the updates. We employ a two-phase algorithm that learns the structure of the game tree and the parameters separately. The algorithm is motivated by the following theorem. Since the result applies broadly to the dual decomposition paradigm, we state the theorem in a slightly more generic form than that required for our purpose. The theorem applies to our setting with

$$f(\mathcal{N}') = -G(\mathcal{N}'), \mathcal{A} = [n], \text{ and } g_i(\Theta_i, N_i) = \sum_{j \neq i} \delta_{ij} N_{ij} - L(\Theta_i, N_i; \delta, \lambda).$$

We now set up the conditions of the theorem. Consider the following combinatorial problem

$$Opt = \max_z \left\{ f(z) + \sum_{\alpha \in \mathcal{A}} g_\alpha(z_\alpha) \right\},$$

where $f(z)$ specifies global constraints on admissible $z$, and $g_\alpha(z_\alpha)$ represent local terms guiding the assignment of values to different subsets of variables $z_\alpha = \{z_j\}_{j \in \alpha}$. Let the problem be minimized with respect to the dual coefficients $\{\delta_{i,\alpha}(z_i)\}$ by following a dual decomposition approach. Suppose we can find a global assignment $\hat{z}$ and dual coefficients such that this assignment nearly attains the local maxima for all $\alpha \in \mathcal{A}$, i.e.,

$$g_\alpha(\hat{z}_\alpha) + \sum_{j \in \alpha} \delta_{j,\alpha}(\hat{z}_j) \geq \max_{z_\alpha} \left\{ g_\alpha(z_\alpha) + \sum_{j \in \alpha} \delta_{j,\alpha}(z_j) \right\} - \epsilon.$$

Assume further, without loss of generality,[1] that the assignment attains the max for the global constraint. Then, we have the following result.

**Theorem 2.** *If there exists an assignment $\hat{z}$ and associated dual coefficients such that the assignment obtains $\epsilon$-maximum of each term in the decomposition, for some $\epsilon > 0$, then the objective value for $\hat{z} \in \big[Opt - |\mathcal{A}|\epsilon, Opt\big]$.*

The theorem implies that if a global structure *nearly* attains the optima for the local neighborhoods, then we might as well shift our focus to finding the global structure rather than optimize for the parameters corresponding to the exact local optima. The result guarantees that the value of such a global structure cannot be too far from that of the optimal global structure.

We outline our two-phase approach in Algorithm 1. The first phase concerns only with iteratively finding a globally consistent structure. It is possible that at the conclusion of this phase, the local structures do not fully agree (the relaxation is not tight). For this reason, the procedure runs for a specified maximum number of iterations and selects the global tree corresponding to an iteration that is least inconsistent with the local neighborhoods. Note that this phase does not precisely solve the original problem we posed earlier. Instead, the structure is obtained without constraining parameters to agree. In this sense, the first phase does not consider strictly potential games as the interactions between players can remain intrinsic to the players themselves.

The second phase simply optimizes the parameters for the already specified global tree. This step realizes a potential game as the parameters and the structure will be in agreement. We note that such parameters could be optimized directly for the selected tree without the need of dual decomposition. However, Algorithm 1 remains suitable in a distributed setting since each player is required to solve only local problems during the entire execution of the algorithm.

### 3.2 Scaling the algorithm

As already noted, Algorithm 1 exhaustively enumerates all neighborhoods for each local optimization problem. This makes the algorithm computationally prohibitive in realistic settings. We now outline an approximation procedure that restricts the candidate neighborhood assignments. Specifically, for a local optimization at any node $i$, we may restrict the possible local neighborhoods at any iteration $t$ to only those configurations that are at most $h$ Hamming distance away from the best local configuration for $i$ in iteration $t$-1. That is, we update each local max-structure incrementally, still guided by the

overall tree within the same dual decomposition framework. Note that we recover Algorithm 1 as a special case when $h = n$. A small $h$ corresponds to searching over a much smaller space compared to the brute force algorithm. For instance, if we take $h = 1$, then the total complexity of the approximate algorithm reduces to $O(n^2 * MaxIter)$ since in each iteration we need to solve $n$ local problems each having $O(n)$ candidate neighborhoods.

---

**Algorithm 1** Learning tree structured potential games

1: **procedure** LEARNTREEPOTENTIALGAME
2: **Input:** parameters $\rho$, $\beta$, $MaxIter$, and $\epsilon > 0$.
3:
4: **Phase 1: Learn Tree Structure**
5:     Initialize $t = 1$, $\lambda_{ij} = 0$, $\delta_{ij} = 0$, $MinGap = \infty$.
6:     **repeat**
7:         Find $\mathcal{N}' = \arg\min_{\mathcal{N}} G(\mathcal{N}, \delta)$ using a minimum spanning tree algorithm
8:         **for** each $i \in [n]$ **do**
9:             **for** each $N_i$ **do**
10:                 Compute $\Theta_i^{*t+1} = \min_{\Theta_i} L(\Theta_i, N_i; \delta, 0)$
11:             Find $N_i^* = \arg\min_{N_i} L(\Theta_i^{*t+1}, N_i; \delta, 0)$
12:             Compute gap: $Gap = \sum_{i,j} \mathbb{I}(N_{ij}^* \neq N_{ij}')$.
13:             **if** $Gap < MinGap$ **then**
                $MinGap = Gap, Global = \mathcal{N}'$
14:             Update $\delta$ $\forall i, j \neq i$: $\delta_{ij} = \delta_{ij} + \beta_t(N_{ij}^* - N_{ij}')$
15:             $t \leftarrow t + 1$
16:     **until** $MinGap = 0$ or $t > MaxIter$.
17:     Set $\mathcal{N}'^* = Global$.
18:
19: **Phase 2: Learn Parameters**
20:     Set $\mathcal{N} = \mathcal{N}'^*$
21:     Compute $\Theta_i^{*t+1} = \min_{\Theta_i} L(\Theta_i, N_i; 0, \lambda)$
22:     **repeat**
23:         Compute $u$ $\forall i, j \neq i$: $u_{ij}^{t+1} = (\theta_{ij}^{*t+1} + \theta_{ji}^{*t+1})/2$
24:         Update $\lambda$ $\forall i, j \neq i$: $\lambda_{ij} = \lambda_{ij} + \rho(\theta_{ij}^{*t+1} - u_{ij}^{t+1})$
25:         Compute $\Theta_i^{*t+1} = \min_{\Theta_i} L_A(\Theta_i, N_i; u_{i\cdot}, 0, \lambda)$
26:         $t \leftarrow t + 1$
27:     **until** $\sum_{i,j \neq i} ||\theta_{ij}^{*t+1} - \theta_{ji}^{*t+1}||_2 < \epsilon$
28:     Set $\theta_{ij}^*, \theta_{ji}^* = (\theta_{ij}^{*t+1} + \theta_{ji}^{*t+1})/2$
29: **Output:** $\mathcal{N}'^*$, $\Theta^*$

---

## 4  Experimental Results

We now describe the results of our experiments on both synthetic and real data to demonstrate the efficacy of our algorithm. We found the algorithm to perform well for a wide range of $C$ and $\beta$ across different data. We report below the results of our experiments with the following setting of parameters: $\rho = 1$, $\beta_t = 0.005$ (for all $t$), $C = 10$, $\epsilon = 0.1$, and $MaxIter = 100$. For each local optimization problem, the configurations were constrained to share the slack variable in order to reduce the total number of optimization variables. Moreover, we used a scaled 0-1 loss [15], $e(y, y^m) = 1\{y \neq y^m\}/n$ for each local optimization. We set $h = 1$ for the approximate method.

We conducted different sets of experiments to underscore the different aspects of our approach. Our experiments with toy synthetic data highlight recovery of an underlying true structure under controlled conditions (pertaining to the data generation process). The results on a real, but toy dataset, Supreme Court vindicate the applicability of the exhaustive approach to unraveling the interactions

latent in real datasets. Finally, we address the scalability issues inherent in the exhaustive search, by demonstrating the approximate version on the larger Congressional Votes real dataset.

## 4.1 Synthetic Dataset

We will now describe how the brute force method recovered the true structure on a synthetic dataset. For this, data were assumed to come from the underlying model

$$f(y; x, \theta) = \sum_{ij \in E} \theta_{ij}(y_i, y_j) + \sum_i x_i \theta_i(y_i),$$

where $x$ represents the context that varies. The parameters were set as follows. We designed a $n$-node degenerate or pathological tree, $n = 6$, with edges between node $i$ and $i + 1$, $i \in \{1, 2, \ldots, n - 1\}$. On each edge $(i, j) \in E$, we sampled $\theta_{ij}(y_i, y_j), y_i, y_j \in \{0, 1\}$ uniformly at random from $[-1, 1]$ independently of the other edges. For each node $i$, we also sampled $\theta_i(y_i), y_i \in \{0, 1\}$ independently from the same range. Each training example pair $(x_m, y_m)$ was sampled in two steps. First, each $x_{mj}, j \in [n]$ was set uniformly at random in the range $[-10, 10]$, independently of each other. The associated $y_m$ was then generated according to the stochastic process described in Section 2. Briefly, starting with $y_m \in \{0, 1\}^n$ sampled uniformly at random, we successively updated the configuration by changing a randomly chosen coordinate of $y_m$, and accepting the move only if the associated score was higher. Since there are $2^n$ possible configurations of binary vectors, we were guaranteed that, in finite time, this procedure ended in a *locally stable* configuration. Once this locally stable configuration was reached, we checked if the score of this configuration exceeded all the other configurations with Hamming distance one by at least $1/n$. If yes, then we included the pair $(x_m, y_m)$ in our synthetic data set, otherwise we discarded the pair. Starting with 100 examples, this procedure resulted in a total of 78 stable configurations that scored higher than each configuration one Hamming distance away by at least $1/n$. These configurations formed our synthetic data set. We were able to exactly recover the tree structure at the end of the Phase 1 of our algorithm using the training data. Fig. 1 shows the evolution of the global tree structure (i.e. $\mathcal{N}'$ in the iterations that resulted in decrease of $Gap$). Note how the algorithm corrects for incorrect edges, starting from a star tree till it recovers the pathological tree structure. Fig. 2 elucidates the synergy between the global tree and local neighborhoods toward recovering the correct structure.

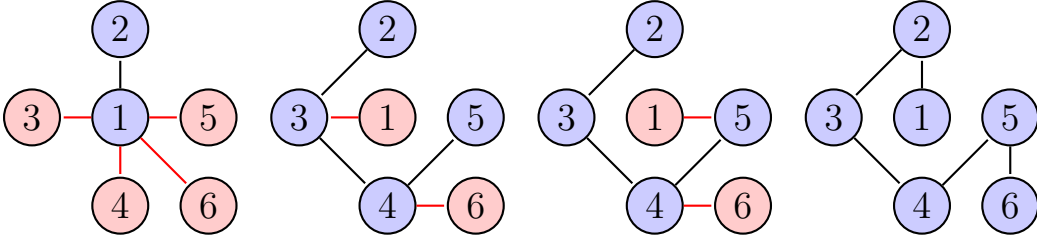

Figure 1: **Recovery on synthetic data**. Evolution of the tree structure is shown from left to right. Each incorrect edge is indicated by coloring one of the end nodes in red. After first iteration, only the edge (1, 2) is identified correctly. At termination, all edges in the underlying structure are recovered.

We show in Fig. 3 the evolution of the tree when the observations were falsely treated as globally optimal points. Clearly, structured prediction failed to recover the underlying tree structure.

## 4.2 Real Dataset 1: Supreme Court Rulings

For both real datasets, we assumed the following decomposition:

$$f(y; \theta) = \sum_{ij \in E} \theta_{ij}(y_i, y_j) + \sum_i \theta_i(y_i).$$

For our first real dataset,[2] we considered the rulings of a Supreme Court bench comprising Justices Alito $(A)$, Breyer $(B)$, Ginsburg $(G)$, Kennedy $(K)$, Roberts $(R)$, Scalia $(S)$, and Thomas $(T)$, during

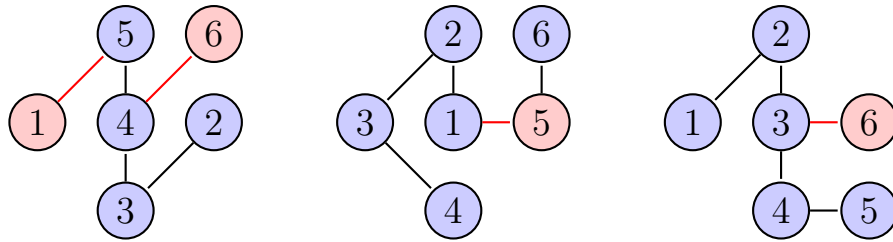

Figure 2: **Global-Local Synergy**. (Center & Right) Spanning trees formed from two separate local neighborhoods (in different iterations). (Left) The common global tree structure. The global tree structure reappears during the execution of the algorithm. On first occurrence, the global tree is misaligned from chain 2-3-4 of the local neighborhood tree at node 5, as indicated by tree in the center. The algorithm takes corrective action, and on the next occurrence, node 5 moves to the desired position, as seen from tree on the right. The algorithm proceeds to correct the positioning of node 6.

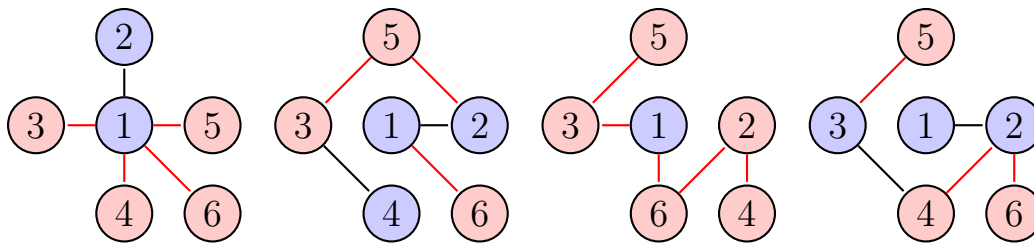

Figure 3: **Evolution of structured prediction**. Structured prediction fails to recover true structure.

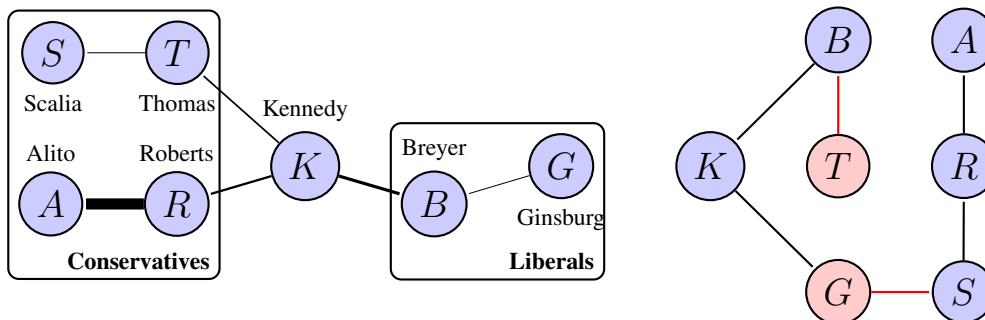

Figure 4: (**Left**) Tree recovered from Supreme Court data. The tree is consistent with widely known ideology of the justices: Justice Kennedy ($K$) is considered largely moderate, while the others espouse a more conservative or liberal jurisprudence. The thickness of an edge indicates the strength of interaction in terms of (scaled) $l_2$-norm of the edge parameters. (**Right**) Enforcing global constraints (structured prediction) resulted in a qualitatively incorrect structure.

the year 2013. Justices Alito, Roberts, Scalia, and Thomas are known to be conservatives, while Justices Breyer and Ginsburg belong to the liberal side of the Court. Justice Kennedy generally takes a moderate stand on most issues. On every case under their jurisdiction, each Justice chose an integer from the set $\{1, 2, \ldots, 8\}$. We considered all the rulings of this bench that had at least one "dissent". For our purposes, we created a dataset from those rulings that did not register a value $6, 7, 8$ from any of the Justices, since these values seem to have a complex interpretation instead of a simple yes/no. For all other values, we used the interpretation by [21]: dissent value 2 was treated as $0$ (no), and others with $1$ (yes). Fig. 4 shows that we were able to recover the known ideology of the Justices by correctly treating the rulings as local optimal, whereas structured prediction failed to identify a qualitatively correct structure.

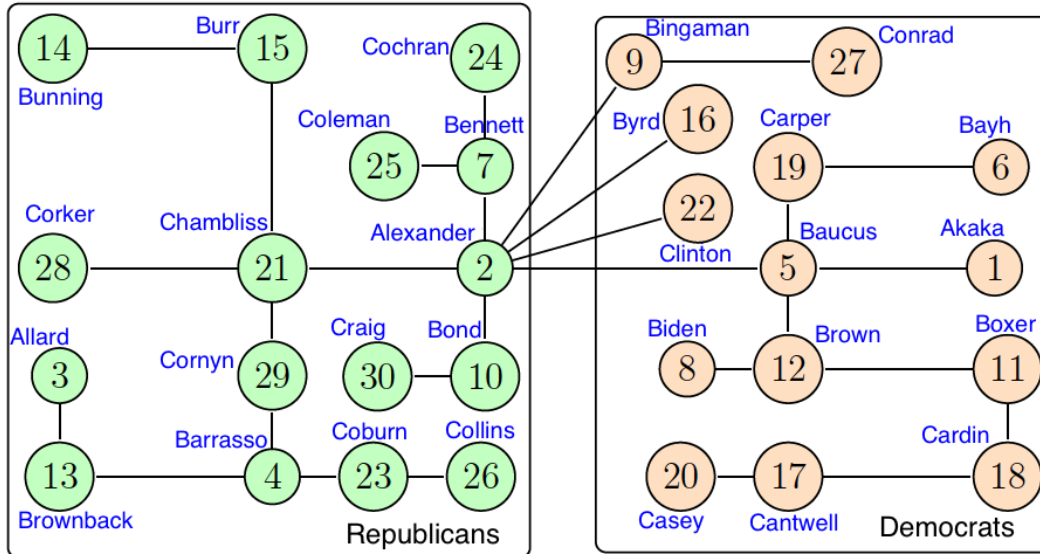

Figure 5: (**Congressional Votes**.) The recovered tree is consistent with the expected voting pattern that, in general, Democrats and Republicans vote along their respective party principles.

## 4.3 Real Dataset 2: Congressional Voting Records

We also experimented with a dataset[3] obtained by compiling the votes on all the bills of the $110^{th}$ United States Congress (Session 2). The US Congress records the voting proceedings of the legislative branch of the US federal government [11]. The U.S. Senate consists of 100 senators: each of the 50 U.S. states is represented by two senators. We compiled all the votes of the first 30 senators (in data order) over this period on bills without unanimity. Each vote takes one of the two values: +1 or -1, to denote whether the vote was in favor or against the proposed bill. We treated vote values -1 with 0 to create a binary dataset. Fig. 5 shows how the approximate algorithm is able to recover a qualitatively correct structure that Democrats and Republicans typically vote along their respective party ideologies (note that there might be more than one qualitatively correct structure). Specifically, we obtain a structure where no Democrat is sandwiched between two Republicans, or vice-versa.

## Discussion

A primary goal of this work is to argue that complex strategic interactions are better modeled as locally optimal solutions instead of globally optimal assignments (as done, for instance, in structured prediction). We believe this local versus global distinction has not been accorded due significance in the literature, and we hope our work fosters more research in that direction.

The work opens up several interesting avenues. All the results presented in this paper are qualitative in nature, primarily because quantitative evaluation is non-trivial in our setting since a strategic game may have multiple equilibria (local optima). The incremental method proposed in this paper does not come with any certificate of optimality, unlike most dual decomposition settings. We assumed the dynamics of the underlying game follow a stochastic process, whereas players typically take deterministic turns in real game settings. From a statistical learning perspective, it will be interesting to estimate the generalization bounds in terms of the number of local equibria samples. Learning across (repeated) games and exploring sub-modular potential functions are other directions.

## Acknowledgments

Jean Honorio provided the Congressional Votes dataset for our experiments. We would also like to thank the anonymous reviewers for their helpful comments.

## Footnotes

[1] We can adjust the bound with a term that depends on the difference between the value of the optimal global structure and the value of the global structure under consideration if these values do not coincide.

[2]Publicly available at `http://scdb.wustl.edu/`.

[3]Publicly available at `http://www.senate.gov/`.

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
