[Supplementary Material]

## Learning Tree Structured Potential Games (Supplementary)

**Theorem 1.** Given a set of training examples $S = \{(x^m, y^m)\}_{m=1}^M$, it is NP-hard to decide whether there exists a tree $T$ and parameters $\theta$ (up to model equivalence) such that

$$\forall\, m, i, y_i, f(y^m; x^m, T, \theta) \geq f(y_{-i}^m, y_i; x^m, T, \theta) + e(y, y^m),$$

when $f(y; x, T, \theta)$ is of the following form:

$$f(y; x, T, \theta) = \sum_{ij \in T} \theta_{ij}(y_i, y_j) + \sum_i \theta_i(y_i) + \sum_i x_i(y_i).$$

*Proof.* Without loss of generality, we omit the margin losses $e(y_i, y_i^m)$ since the bias terms $x_i(y_i)$ suffice to rule out the trivial solution $\theta = 0$. We prove the result via a poly-time reduction from the bounded-degree spanning tree problem. The decision version of the bounded-degree spanning tree problem is stated as follows. Given an undirected graph $G$, does there exist a spanning tree with maximum degree $D$ in $G$? This problem is known to be NP-hard [17].

We construct an instance of our problem from an instance of the bounded-degree spanning tree problem in the following way. Let $n = |V|$ be the number of vertices in the input graph $G = (V, E)$. We define variables $y_1, y_2, \ldots, y_n$, where each $y_i \in \{1, 2, \ldots, n+1\}$. Let $deg_T(i)$ be the degree of node $i \in \{1, \ldots, n\}$ in $T$. Next we define a set of parameters $\theta$ as

$$\theta_i(y_i) = \begin{cases} D & \text{if } y_i = n+1 \\ 0 & \text{otherwise} \end{cases}, \text{ and}$$

$$\theta_{ij}(y_i, y_j) = \begin{cases} 1 & \text{if } y_i \neq n+1 \text{ and } y_j = n+1, \text{ or } y_j \neq n+1 \text{ and } y_i = n+1 \\ 0 & \text{if } y_i = y_j = n+1 \\ -n^2 & \text{otherwise.} \end{cases}$$

We now construct a training set that we would like to be separable only by the desired parameters. Let $z_i(j)$ be the $n$-dimensional vector $(n+1, \ldots, \underbrace{j}_{\text{index } i}, \ldots, n+1)$ with all entries identically set

$$\underbrace{\qquad\qquad\qquad\qquad\qquad\qquad}_{n-\text{component vector}}$$

to $n+1$, other than index $i \in \{1, \ldots, n\}$ with entry $j \in \{1, 2, \ldots, n\}$. Construct a set of $O(n^2)$ examples $S = \{(0, 0, \ldots, 0), z_i(j)\}_{i \in \{1, \ldots, n\}, j \in \{1, \ldots, n\}}$. Consider $(x = (0, \ldots, 0), \tilde{y}) \in S$, where $\tilde{y} = (n+1, \ldots, n+1)$ is a labeling vector with each entry set to $n+1$. Note that the bias terms $x_i(\tilde{y}_i)$ are identically 0 for all examples in $S$. For this example, we have

$$f(\tilde{y}; (0, \ldots, 0), T, \theta) = nD,$$

since the pairwise $\theta$ values are all zero, and only non-zero contribution comes from $\sum_i \theta_i(\tilde{y}_i)$. Every other assignments $y$ is of the form $y = z_i(j)$, where $i, j \in \{1, \ldots, n\}$. For any $i, j \in \{1, \ldots, n\}$, we have the contribution of singleton potential to $f(y; (0, \ldots, 0), T, \theta) = f(z_i(j); (0, \ldots, 0), T, \theta)$ is

$$\sum_k \theta_k(y_k) = \underbrace{\theta_i(y_i)}_{=0} + \sum_{k \neq i} \underbrace{\theta_k(y_k)}_{=D} = (n-1)D.$$

On the other hand, for all $i$, and $j$, the contribution of the pairwise potentials to $f(y; (0, \ldots, 0), T, \theta)$ is simply $deg_T(i)$, the number of neighbors of node $i$ in $T$. Therefore, we have,

$$f(y; (0, \ldots, 0), T, \theta) = f(z_i(j); (0, \ldots, 0), T, \theta) = (n-1)D + deg_T(i)$$

Since the assignments $\tilde{y}$ and $y$ differ in exactly one component, we have that $S$ is separable by $\theta$ if and only if there exists a spanning tree $T$ such that

$$f(\tilde{y}; (0, \ldots, 0), T, \theta) \geq f(y; (0, \ldots, 0), T, \theta),$$

or equivalently iff

$$nD \geq (n-1)D + deg_T(i), \forall\, i,$$

i.e., iff $D \geq deg_T(i) \,\forall\, i$, which is just the bounded degree spanning tree problem. Thus we have shown that $S$ is separable by $\theta$ if and only if there exists a spanning tree $T$ of degree at most $D$. i.e. if and only if the answer to the bounded degree spanning tree problem is "yes".

To complete the proof, we need to recover $\theta$. To this end, we can express $\theta$ in its equivalent canonical form with respect to assignment $(n+1, n+1, \ldots, n+1)$ using the procedure outlined in [15]. Then, we can construct a set $S'$ of $M = 2|V|n + 2|E|n^2 = 2n^2(1 + |E|)$ examples $\{x^m, y(x^m, \theta)\}$ using the procedure in Proposition 2.1 of [15] to recover (canonical) $\theta$. This procedure is polynomial, and ensures that every example satisfies the following property: for each $i \in \{1, \ldots, n\}$, either the label $y_i^m$ or its neighbors are set to the canonical state $n+1$. Note that set $S$ described above also satisfies this property. Clearly, the entire training set $S \bigcup S'$ is of size $O\left((1 + |E|)n^2\right)$, which is polynomial in $n$. Thus we have outlined a procedure to recover $\theta$ in polynomial time, and the proof is complete. $\square$

**Theorem 2.** Consider the following combinatorial problem

$$Opt = \max_z \left\{ f(z) + \sum_{\alpha \in \mathcal{A}} g_\alpha(z_\alpha) \right\}, \tag{2}$$

where $f(z)$ specifies global constraints on admissible $z$, and $g_\alpha(z_\alpha)$ represent local terms guiding the assignment of values to different subsets of variables $z_\alpha = \{z_j\}_{j \in \alpha}$. Let the problem be minimized with respect to the dual coefficients $\{\delta_{i,\alpha}(z_i)\}$ by following a dual decomposition approach. Suppose we can find a global assignment $\hat{z}$ and dual coefficients such that this assignment nearly attains the local maxima, i.e.,

$$g_\alpha(\hat{z}_\alpha) + \sum_{j \in \alpha} \delta_{j,\alpha}(\hat{z}_j) \geq \max_{z_\alpha} \left\{ g_\alpha(z_\alpha) + \sum_{j \in \alpha} \delta_{j,\alpha}(z_j) \right\} - \epsilon \tag{3}$$

for some $\epsilon > 0$, across the components $\alpha \in \mathcal{A}$. Assume further, without loss of generality,[4] that the assignment attains the max for the global constraint. Then, we have that

$$\max_z \left\{ f(z) - \sum_{\alpha, i \in \alpha} \delta_{i,\alpha}(z_i) \right\} + \sum_{\alpha \in \mathcal{A}} \left[ \max_{z_\alpha} \left\{ g_\alpha(z_\alpha) + \sum_{j \in \alpha} \delta_{j,\alpha}(z_j) \right\} - \epsilon \right] \in \left[Opt - |\mathcal{A}|\epsilon, Opt\right].$$

*Proof.* Typically, if the local maximizing assignments are consistent across the components for some choice of dual coefficients, then the resulting assignment is optimal, and the upper bound is tight. However, under the conditions of the theorem, we have

$$\max_z \left\{ f(z) + \sum_{\alpha \in \mathcal{A}} g_\alpha(z_\alpha) \right\} \tag{4}$$

$$\geq\; f(\hat{z}) + \sum_{\alpha \in \mathcal{A}} g_\alpha(\hat{z}_\alpha) \tag{5}$$

$$=\; f(\hat{z}) - \sum_{\alpha, i \in \alpha} \delta_{i,\alpha}(\hat{z}_i) + \sum_{\alpha \in \mathcal{A}} \left\{ g_\alpha(\hat{z}_\alpha) + \sum_{j \in \alpha} \delta_{j,\alpha}(\hat{z}_j) \right\} \tag{6}$$

$$\geq\; \max_z \left\{ f(z) - \sum_{\alpha, i \in \alpha} \delta_{i,\alpha}(z_i) \right\} + \sum_{\alpha \in \mathcal{A}} \left[ \max_{z_\alpha} \left\{ g_\alpha(z_\alpha) + \sum_{j \in \alpha} \delta_{j,\alpha}(z_j) \right\} - \epsilon \right] \tag{7}$$

$$\geq\; \max_z \left\{ f(z) + \sum_{\alpha \in \mathcal{A}} g_\alpha(z_\alpha) \right\} - |\mathcal{A}|\epsilon. \tag{8}$$

In other words, such consistent $\hat{z}$ is $|\mathcal{A}|\epsilon$ optimal even though it does not attain local maxima for the components. The goal can be therefore slightly shifted towards finding consistent assignments at the expense of obtaining exact local maxima. $\square$

## Footnotes

[4]We can adjust the bound with a term that depends on the difference between the value of the optimal global structure and the value of the global structure under consideration if these values do not coincide.