[Reviews · NeurIPS 2016]

Reviewer 1

Summary

This paper investigates the problem of learning potential games that are restricted to tree structures. This is an interesting problem since it is natural to explain complex data as the result of such games. A maximum margin approach is adopted that creates a difficult optimization problem, so a dual decomposition approach is introduced instead. Promising experiments on synthetic and opinion/voting data are provided that show improved predictive accuracy (synthetic data) and sensible tree recovery (real data).

Qualitative Assessment

Overall, the paper is well written and motivated clearly. Game formulations are nicely incorporated into the prediction task while building upon the relevant structured prediction literature. The problem area is important, but somewhat niche. The paper first proves that max margin learning of tree structured models is NP-hard. Though I am fairly convinced that this problem is NP-hard, the proof should be clarified. The theorem statement allows both T and \theta to be chosen, while the proof fixes \theta and shows that finding T would solve the NP-hard max degree spanning tree problem. Results for the recoverability of \theta are included at the end, but it is unclear how these prevent other choices of \theta from potentially satisfying the inequalities. Additionally, I would expect the pairwise potentials to be based on the input graph, G, since the theorem statement does not include an “input” graph in which the tree is constrained. All of the experiments (even synthetic) are on small problems with at most n=8 variables. Especially for the synthetic experiments, how does the run time performance scale as a function of n? For the real datasets, there are many simpler methods for constructing such spanning trees. Can anything be said about how well the method improves predictions of voting similarity?

Confidence in this Review

2-Confident (read it all; understood it all reasonably well)


Reviewer 2

Summary

This paper considers a tree-structured potential game its associated learning problem. They showed NP-hardness of the learning problem. On the other hand, they propose an algorithm that finds a local optimum of the objective function. In the experiments, the proposed algorithm outperforms standard structured prediction baselines on artificial and real data sets.

Qualitative Assessment

This paper considers an interesting learning problem that might motivates further study. On the other hand, the theoretical analysis looks somewhat standard such as NP-hardness and lacks analysis of generalization performance.

Confidence in this Review

1-Less confident (might not have understood significant parts)


Reviewer 3

Summary

The paper incorporates dual decomposition methods into "potential games / graphical games". Such games are played among many players and and their joint strategy is described by a single potential function, which is described using sparse payoffs (graphs). An equilibrium of such a game is all players do not change their play given the state of all players. This paper considers payoffs structures that form a tree. The authors show that it is NP-hard to recover tree-structured games and suggest an dual decomposition approach to learn such games.

Qualitative Assessment

The suggested approach extends the reach of structured prediction beyond the standard "find the maximal assignment" which has been extensively studied in the last decade. It is a refreshing direction - handling Nash equilibrium using structured models may have a significant impact in the future. The authors also prove that the problem is hard. While it is nice to know we are dealing with hard problems, I am personally more excited by the dual decomposition approach which seems to work well. The authors solve the dual program by ADMM, a method that gained popularity in the last years. Lastly, in a series of illuminating experiments, both on synthetic and real data, the authors show the benefit of their approach.

Confidence in this Review

2-Confident (read it all; understood it all reasonably well)


Reviewer 4

Summary

The paper presents a method for learning the tree structure and parameters of potential games by constructing a max-margin loss for locally-optimal assignments. The authors show that the learning objective is NP-hard, and propose an approximate dual-decomposition optimization technique. Some experiments on (very) small graphs are presented.

Qualitative Assessment

The problem presented in the paper is interesting. In general the notion of using games and their results as samples is intriguing. The locally-optimal nature of equilibria seems interesting to explore, especially since many methods (such as structured prediction) usually focus on MAP assignments. While the general idea is interesting, the experiments are disappointing, and suggest that the method is computationally prohibitive and cannot be used in practice. The lack of guarantees on the optimization process is also discouraging. Remarks: 1) The formulation of the locally-optimal max-margin constraints is an interesting notion. It is not clear however why the authors believe that a discriminative method should return the 'correct' tree and/or parameters, as such an approach is oriented towards optimal *predictions*. 2) Since the stochastic model generates a distribution over y given x, it is not clear how predictions are made using a learned model, if at all. Moreover, the loss punishes a 'prediction' for being different than an observed label, while in truth it is only one of many possible legitimate labels. This seems unnecessarily harsh. 3) While the dual-decomposition approach looks promising, a downside is that there are no convergence guarantees on the coordinate-descent-style optimization procedure. 4) The experiments are rather disappointing. Why are they over graphs with at most 8 (!) nodes? Why are only chain graphs used as ground truth, and not random trees? Why are 100 iterations necessary for convergence? If runtime is a drawback, this should be mentioned and theoretically analyzed. This hints that the method is far from applicable. 5) The authors mention that the accuracy of their method is better, though it is unclear what exactly is measured. On the one hand, the authors claim that the goal is to reconstruct the base tree, but in all samples share the same tree. On the other hand, if accuracy relates to predictions, it is not clear (a) how they are made, and (b) how they are measured w.r.t the the probabilistic labels in the test set. 6) It is not clear why the authors believe that the real data used is based upon a tree and not a general graph. This makes the task of finding the 'true' tree puzzling. Separating conservatives from liberals is an easy task for any clustering algorithm.

Confidence in this Review

2-Confident (read it all; understood it all reasonably well)


Reviewer 5

Summary

This paper introduces a max margin framework for learning potential games where the underlying interactions have a tree structure. A hardness result is presented first, followed by a dual decomposition algorithm for learning the tree structure and parameters of the game. Lastly, experimental results on synthetic and real datasets show that the algorithm recovers reasonable results.

Qualitative Assessment

I like the underlying idea of this paper, that of applying max margin learning to potential games. This works well with viewing observed behavior as local equilibria of the potential function. However, I have a couple serious concerns about the execution and technical content. My major concern is that the paper very quickly glosses over the issue of learning N_i, the set of neighbors of each node. This appears to be the key part of the inference: as the authors point out, finding the optimal parameters with the tree structure fixed is just a quadratic program. It is claimed that special properties like submodularity could be used, but no such cases are explored in the paper. This leaves brute force, which is clearly not sufficient to tackle realistically sized problems. Also, it is never mentioned which method is used in the experiments. Next, it would be helpful to see more detailed experiments. The synthetic and real datasets are good for illustrating that the algorithm is on some level reasonable, but two dimensions are somewhat lacking. First, all of the datasets considered are very small (and no runtime results are shown). So, it's hard to know how the algorithm scales, or whether it gives good results outside of small, clearly structured examples. Second, there is no comparison to other techniques for learning games. I'm not able to give a comprehensive list of recommendations, but a couple of papers which tackle similar problems are Quang Duong, Yevgeniy Vorobeychik, Satinder Singh, Michael P. Wellman. "Learning Graphical Game Models". IJCAI 2009. Jean Honorio, Luis Ortiz. "Learning the Structure and Parameters of Large-Population Graphical Games from Behavioral Data". UAI 2012. Some comparison to alternative ways of learning graphical games would be serve to validate the overall max margin approach. A few smaller issues: 1) It's not clear where the assumed tree structure makes the inference easier. I assume that it simplifies learning the parameters, but this should be explicitly discussed somewhere. 2) The meaning of the "context" is not clear. What was the context in the two real world datasets? 3) In the equation just before line 59, "y" is undefined. 4) In the Lagrangian (just after line 118), is delta_ij the Kronecker delta? If so, isn't the summation identically zero?

Confidence in this Review

2-Confident (read it all; understood it all reasonably well)


Reviewer 6

Summary

The paper considers tree structured potential games where the payoff of each player depends only on their own action and their neighbors' actions. It shows that it is NP-hard to learn the tree structure and associated parameters of such games under a max-margin setting. It proposes a dual decomposition method to solve the problem, and give some interesting examples.

Qualitative Assessment

The proposed setting seems interesting and applicable to problems in real life. The experiments are interesting, and give somewhat reasonable results. There is one question though, how can the authors be sure that the expected voting pattern is in fact correct? For example, in the Congressional Voting Records example, how can it be verified that there are no extra edges between senators from different parties? (There could be unknown connections, which the authors do not know, and not discovered by the model). It seems that the result of the paper can only be critically examined if the real relations between different agents are known.

Confidence in this Review

1-Less confident (might not have understood significant parts)